# Antimicrobial Resistance in Companion Animals: A 30-Month Analysis on Clinical Isolates from Urinary Tract Infections in a Veterinary Hospital

**DOI:** 10.3390/ani15111547

**Published:** 2025-05-25

**Authors:** Raffaele Scarpellini, Silvia Piva, Erika Monari, Kateryna Vasylyeva, Elisabetta Mondo, Erika Esposito, Fabio Tumietto, Francesco Dondi

**Affiliations:** 1Department of Veterinary Medical Sciences, Alma Mater Studiorum—University of Bologna, Via Tolara di Sopra n 50, 40064 Ozzano dell’Emilia, Italy; raffaele.scarpellini@unibo.it (R.S.); silvia.piva@unibo.it (S.P.); kateryna.vasylyeva2@unibo.it (K.V.); elisabetta.mondo2@unibo.it (E.M.); erika.esposito6@unibo.it (E.E.); f.dondi@unibo.it (F.D.); 2Unit of Antimicrobial Stewardship, Local Health Authority of Bologna, 40138 Bologna, Italy; fabio.tumietto@ausl.bologna.it

**Keywords:** companion animals, multidrug antibiotic resistance, surveillance, small animal infection, cystitis, subclinical bacteriuria, ISCAID guidelines

## Abstract

Antimicrobial resistance is a frequent finding during bacterial urinary tract infection in dogs and cats. This study evaluated bacterial species, antimicrobial resistance patterns and multi-drug resistance rate in clinical isolates of dogs and cats in a European veterinary university hospital; moreover, we evaluated multi-drug resistance trend in a long-term follow-up, applying the International Society for Companion Animal Infectious Disease urinary tract infections guidelines as a guide for antimicrobial stewardship. In our study, almost half of the cases were classified as upper urinary tract infection and recurrent cystitis; also, this study showed that antimicrobial resistance reached 75% of clinical isolates and the multi-drug resistance percentage was 37%, including resistance mostly against first-line antibiotics such as penicillins for sporadic cystitis and fluoroquinolones for pyelonephritis, respectively. Applying urinary tract infection guidelines during a 30-month period of follow-up, led to a significant decrease in multi-drug resistance. Our work highlights how the application of international urinary tract infection guidelines as a tool for antimicrobial stewardship can significantly reduce antimicrobial resistance in daily practice for a long-term follow-up; moreover, it emphasizes the bacteriological assessment as a fundamental examination for urinary tract infection treatment, guiding antibiotics prescription in an evidence-based manner to reduce antimicrobial resistance spreading.

## 1. Introduction

The spread of antimicrobial resistance (AMR) is a major threat for both human and animal health worldwide. The selective pressure exerted by antimicrobial use (including misuse and overuse) is considered one of the most important factors for an AMR rise [1]. In veterinary medicine, companion animals are gaining interest regarding their role in the overall AMR epidemiology. Indeed, although few evidence-based data are available to date, their close proximity with people could facilitate the interspecies transmission of AMR, and poses a significant One Health challenge [2]. Moreover, infections in dogs and cats are mostly treated with the same antibiotics used in human medicine, such as fluoroquinolones, penicillins and cephalosporins [3], increasing the risk of development of the same drug resistances. Specifically, urinary tract infections (UTIs) are commonly encountered in veterinary practice [4,5,6,7], affecting a substantial proportion of companion animals each year. It has been estimated that close to 14% of dogs evaluated by a veterinarian will develop UTI during their lifetime, with the average age being 7.8–8 years [8,9]. Therefore, similarly to human medicine [10,11], UTIs are also a frequent reason for antimicrobial prescription in small animal practices [3,4,12,13].

While antimicrobial therapy has traditionally been the cornerstone for UTI treatment, the emergence of multidrug-resistant (MDR) pathogens has complicated management strategies, raising serious public health implications. UTIs represent a significant source of morbidity among companion animals, leading to clinical signs of different severity, a decreased quality of life and potential life-threatening complications if left untreated. Additionally, the increasing involvement of AMR bacteria is also paramount for humans they cohabit with. Moreover, multi-drug resistance is exacerbated by factors such as inappropriate antimicrobial selection, suboptimal dosing and the difficulties to establish robust antimicrobial stewardship programs in veterinary settings [14,15]. In 2019, the International Society for Companion Animals Infectious Disease (ISCAID) redacted specific guidelines for UTI treatment in companion animals [16]; notably, some of the antibiotics suggested for UTI treatment in dogs and cats are also used in human medicine, and some of them, quinolones and third-generation cephalosporins, specifically, are considered of highest priority and of critically importance (HPCIAs) by the World Health Organization [17]. These guidelines are general recommendations, requiring a proper adjustment considering national legislation on antimicrobial prescriptions, specific geographical AMR rates and epidemiological nuances [16]. In Italy, few recent studies that focused on this specific topic highlighted overall MDR rates up to 44% [18] and 60% in animals previously treated [19], with alarming resistance rates for HPCIAs such as fluoroquinolones (36%) and third-generation cephalosporins (29.7%) in uropathogenic *Escherichia coli* [20]. Given these premises, there is an urgent need for comprehensive research addressing AMR in UTIs among companion animals, including local data about etiology, AMR rates and their relationship with antibiotics administration [21].

This study had different aims: (1) to describe the prevalence of bacteria isolated from samples collected in dogs and cats with UTI evaluated at a Veterinary University Hospital (VUH); (2) to determine non-susceptibility patterns of such isolates; (3) to describe antimicrobial usage patterns in positive specimens; (4) to evaluate the impact of the application of the ISCAID guidelines on both AMR rates and antimicrobial usage over a 30-month period.

## 2. Materials and Methods

A prospective observational study, part of a larger surveillance program whose partial results were published in 2023 [22] was conducted at the Veterinary laboratory of bacteriology and at the VUH of the Department of Veterinary Medical Sciences (University of Bologna), from December 2020 to May 2023. This program included a specific part for the application of the ISCAID guidelines [11] for the diagnosis, classification, and management of UTIs in dogs and cats. The implementation of the ISCAID guidelines was preceded and facilitated by specific internal training meetings within the medical staff of the veterinary hospital and subsequently supervised by clinicians of the nephrology and urology unit. Specifically, around 70% of the cases included in this study were followed up directly by the clinicians in this unit, and for an additional 20%, a specific consultation was required for UTI treatment. Additionally, another part of the surveillance program was constituted by meetings between the bacteriology unit and the clinicians, where data collected were systematically shared and discussed.

Samples collected at the VUH from dogs and cats with suspected UTIs were submitted for bacteriological diagnostic purposes and results were recorded and analyzed. Standard microbiological procedures were used and are listed in Table A1. Specimens were classified as urine, bladder stones and urinary bladder biopsies. After an incubation phase of 24–48 h at 37 ± 1 °C, plates with adequate bacterial growth were considered positive according to previous published guidelines [23]. Colonies were morphologically evaluated, and the identification of bacterial species was assessed using the matrix-assisted laser desorption–ionization time-of-flight mass spectrometry method (MALDI-TOF MS) (Biotyper, Bruker Daltonics, Billerica, MA, USA), following manufacturer’s instructions (Bruker Daltonik, Bremen, Germany). With this method, an ID score > 2 (green—high accuracy) or >1.8 (for *Staphylococcus* spp. isolates) was defined as positive for species identification. If the same bacterial species were isolated from different specimens from the same patient at the same time, they were considered as one.

According to the ISCAID guidelines [11] and based on the signalment, history, clinical signs and clinicopathological and imaging findings of the included patients, each positive sample was classified as sporadic bacterial cystitis (SBC), recurrent bacterial cystitis (RBC), upper urinary tract infection (uUTI), bacterial prostatitis (BP), subclinical bacteriuria (SUB) or catheter-associated urinary tract infection (CAUTI). Data about previous hospitalization/surgery in the past 30 days (yes/no), hospitalization at the time of sampling (yes/no), hospitalization in intensive care unit (yes/no) and surgery at the time of sampling (yes/no) were recorded in addition to medical data above reported. Moreover, antimicrobial use in the past 90 days (yes/no, number of antimicrobials, type of drugs used) and current antimicrobial use (yes/no, number of antimicrobials, type of drugs used) were also registered.

Antimicrobial susceptibility testing (AST) of all isolates was performed using the Kirby–Bauer disc diffusion method, according to the Clinical and Laboratory Standard Institute (CLSI) guidelines [24]. Overall, 12 antimicrobials from eight antimicrobial classes were included in the final analysis (Table A2 in Appendix A). All the discs were purchased from a commercial supplier (Oxoid, Milan, Italy). For every tested drug, each isolate was classified as susceptible (S), intermediate (I), or resistant (R) based on the 2020 CLSI veterinary breakpoints or, when not specifically present, human ones [25]. Subsequently, they were updated according to the 2024 version of the document [26]. For Gram-negative bacteria, clindamycin and erythromycin were not tested due to their known low-activity rates [27]. For some species, antimicrobials known to exhibit expected resistance phenotypes (intrinsic resistance), according to the National Reference Laboratory for AMR [28], were not tested and excluded from the analysis. Isolates identified as the same bacterial species, in the same patient, with the same AST profile results at different time points were considered as duplicates, and only the first one chronologically identified was included in the study. For AST interpretation, the strains were divided into “susceptible” and “non-susceptible,” as previously suggested by Sweeney et al. [29], where the “non-susceptible” category included resistant and intermediate isolates. Isolates that were non-susceptible to at least one antimicrobial drug were considered as AMR isolates, whereas isolates that were not susceptible to at least one antimicrobial drug in three or more antimicrobial classes were considered as MDR, according with the definition given by Magiorakos et al. [30].

Descriptive statistics were performed to evaluate patients’ clinical data, type of specimen, ISCAID UTI classification, patients’ hospitalization data, bacterial species identified, mixed or single-species infection, previous and current antimicrobial use, non-susceptibility percentages towards each tested drug and AMR/MDR percentages. Data regarding age were presented as median and range, while all the non-susceptibility percentages were shown with the 95% confidence interval (CI). Differences between dogs and cats in terms of prevalence of bacterial species, non-susceptibility rates and antibiotic usage patterns were statistically evaluated using the Fisher exact test. In a similar way, considering a previous manuscript by Marques et al. [7], a statistical analysis of temporal trends of MDR percentage and single non-susceptibility was performed with a logistic regression (stepwise approach) considering the semester of isolation (from 1st to 5th) as a continuous variable. The same statistical analysis was used to describe the temporal trend for previous and current antimicrobial use. Odds ratios (ORs) were calculated considering the baseline category the first semester of observation. The associations between MDR isolates and the antimicrobial drug previously used in both dogs and cats were calculated using a univariable logistic regression analysis. Statistically significant results were included in the multivariate analysis model, built up with a stepwise selection. A *p* value < 0.05 was used to determine statistically significant results. The normality and heteroskedasticity of data were assessed with the Shapiro–Wilk test and Levene’s test. A statistical analysis was performed with the commercially available MedCalc^®^ statistical software package version 22.009 (MedCalc Software Ltd., Ostend, Belgium).

## 3. Results

### 3.1. Bacterial Identification and Data Collection Results

From 2049 specimens collected from 1311 patients, 670/2049 (32.7%) samples were positive for bacterial growth from 495/1311 patients (37.8%). A total of 729 isolates were obtained, including 602 (82.6%) from 555 specimens of 411 dogs (182 males and 229 females), and 127 isolates (17.3%) from 116 specimens of 84 cats (37 males and 47 females). The age distribution of sampled patients ranged from 1 year (≤1 year) to 17 years in dogs and from 1 year (≤1 year) to 19 years in cats, with a median age of 10 years for both species (interquartile range: 6 years).

The most frequently isolated bacterial species (Table 1) was *Escherichia coli* in both dogs (52.8%) and cats (45.7%). *Staphylococcus felis* was found only in cat specimens (5.5%). In cats, a higher proportion of *Enterococcus faecalis* (15%), compared with dogs (5.5%, *p* < 0.001) was recorded. Significant higher proportions in cats were also recorded for *Corynebacterium urealyticum* (*p* = 0.039), *Staphylococcus aureus* (*p* = 0.039) and Coagulase-negative *Staphylococci* (CoNS), other than *Staphylococcus felis* (*p* = 0.010). On the other hand, *Streptococcus canis* was isolated more frequently in dogs (5.8%) than in cats (0.8%, *p* = 0.012).

Considering the 670 specimens, 662 (98.8%) were urine samples collected by cystocentesis, 5 (0.7%) were urinary bladder biopsies, 1 (0.2%) was a portion of an ureteral stent and 2 (0.3%) were bladder stones collected during endoscopy or surgical procedures. In 612 specimens (91.3%), bacteria were found in monoculture, while in 58 specimens (8.7%), mixed infections were identified, including 57 with two bacterial species and one with three species. The distribution of patients’ hospitalization data is shown in Table 2. Overall, 25% of the patients were hospitalized when specimens were sampled.

Considering only dogs, 39.5% (*n* = 219) of the submitted specimens reported an antimicrobial use in the previous 90 days; specifically, 89.5% (*n* = 196) were specimens with a previous one-drug treatment, 8.7% (*n* = 19) with a two-drug treatment and 1.8% (*n* = 4) with a three-drug administration, respectively. The most used antimicrobial was amoxicillin-clavulanate (Figure 1).

Additionally, 13.2% of these previously cited specimens (*n* = 73) were collected from patients under antibiotic treatment at the time of sampling, including 66 specimens (90.4%) with one drug reported and 7 (9.6%) with double antibiotic treatment, respectively. Marbofloxacin was the most used antibiotic (Figure 2) at the time of specimen sampling.

In cats, similar percentages were recorded, with 39.7% (*n* = 46) of the total specimens obtained from animals previously treated with antimicrobials; specifically, 39 (84.8%) were treated with one antibiotic, 6 (13.0%) with two antibiotics, and 1 (2.2%) with three antibiotics. Amoxicillin-clavulanate represented the most previously used drug (Figure 1). Considering specimens from cats under antibiotic treatment at the time of sampling (*n* = 19, 16.4%), 16 (84.2%) were under treatment with one antibiotic, while 3 specimens (15.8%) reported a two-antibiotic administration, respectively. Again, marbofloxacin was the most used antibiotic at the time of specimen sampling (Figure 2). No significant differences were recorded in the antimicrobial drugs used between dogs and cats.

Considering ISCAID UTI classification, specimens from both dogs and cats were classified based on the type of UTI as follows: 190 (28.4%) SBC, 166 (24.9%) RBC, 167 (24.9%) uUTI, 114 (17%) as SUB, 16 (2.4%) as BP and 12 (1.8%) as CAUTI, respectively. Five cases (0.7%) were not classifiable. No significant differences in incidence of infections were recorded between dogs and cats (Table 3).

Recurrent bacterial cystitis was the most prevalent ISCAID type of UTI among mixed infections (13.8%), while CAUTI (*n* = 8, 66.7%) and RBC (*n* = 110, 65.9%) were the types of infections with the highest frequency with previous antibiotic use; CAUTI (*n* = 5, 41.7%) and uUTI cases (*n* = 38, 22.8%) were the type of UTIs with the highest proportion regarding specimens for which the antibiotics were administered at the time of sampling.

### 3.2. AST Results

Overall, the AMR percentage was 75.5% (*n* = 550, 95% CI 71.8 to 78.1), while MDR percentage was 37.3% (*n* = 272, 95% CI 33.5 to 40.1). In dogs, the MDR percentage was 37.2% (*n* = 224, 95% CI 33.2 to 40.8), while in cats, it was 37.8% (*n* = 48, 95% CI 29.7 to 46.3). The non-susceptibility percentages for the tested drugs are shown in Table 4.

In dogs, higher rates were recorded for clindamycin (65.1%), erythromycin (61.5%), ampicillin (54.5%) and enrofloxacin (44.4%); in cats, clindamycin (53.5%), enrofloxacin (50%), tetracycline (45%) and ampicillin (44.4%) were the antimicrobials tested with more associated non-susceptibility. Compared with cats, in dogs, a statistically significant higher non-susceptibility percentage was recorded for ampicillin (*p* = 0.048); on the other hand, in cats, a statistically significant higher non-susceptibility percentage for ceftiofur was recorded (*p* = 0.028).

Distribution of MDR isolates considering bacterial species is shown in Table 5. *E. coli* (44.1%) and *Staphylococci* part of the SIG (18.4%) were the most frequent MDR species. Additionally, *Enterococcus faecium* (80%) and the SIG members (74.6%) were the bacterial species with the highest MDR proportion within the species.

### 3.3. Temporal Trend and Multivariate Analysis Results

Temporal trend analysis for non-susceptibility rates showed a significant decrease during the five semesters regarding MDR percentage (Figure 3) (*p* = 0.001, OR 0.811, 95% CI 0.728 to 0.904) and MDR *E coli* percentage (*p* = 0.013, OR 0.824, 95% CI 0.707 to 0.960. Additionally, significant decreases were registered for non-susceptibility percentages towards amoxicillin-clavulanate (*p* < 0.001, OR 0.763, 95% CI 0.671 to 0.869), amikacin (*p* = 0.004, OR 0.712, 95% CI 0.588 to 0.862), gentamicin (*p* = 0.001, OR 0.756, 95% CI 0.655 to 0.874), piperacillin-tazobactam (*p* = 0.002, OR 0.772, 95% CI 0.671 to 0.888), cephazolin/cephalothin (*p* = 0.019, OR 0.861, 95% CI 0.759 to 0.976), tetracycline (*p* = 0.005, OR 0.858, 95% CI 0.770 to 0.956), enrofloxacin (*p* = 0.002, OR 0.850, 95% CI 0.766 to 0.944) and trimethoprim-sulfamethoxazole (*p* < 0.001, OR 0.787, 95% CI 0.687 to 0.902).

A temporal trend analysis for antimicrobial use in patients with positive specimens included in the study (Figure 3) showed a significant decrease in the frequency of overall antimicrobial administration at the moment of sampling (*p* < 0.001, OR 0.752, 95% CI 0.641 to 0.882) considering the frequency of overall previous treatment (*p* = 0.018, OR 0.876, 95% CI 0.784 to 0.9775) and the previous use of marbofloxacin (*p* = 0.002, OR 0.700, 95% CI 0.555 to 0.882) and piperacillin-tazobactam (*p* = 0.013, OR 0.563, 95% CI 0.360 to 0.887).

The multivariate analysis between MDR and previous antimicrobial use showed that isolates from patients previously treated with amoxicillin-clavulanate (*p* = 0.001), marbofloxacin (*p* < 0.001), enrofloxacin (*p* < 0.001) and piperacillin-tazobactam (*p* = 0.015) were significantly associated with higher MDR rates.

## 4. Discussion

In our study, we analyzed specimens from dogs and cats with UTIs evaluated at an Italian VUH to obtain a complete overview on local epidemiological data and to better explore the relationship between antimicrobial use and the onset of antimicrobial resistance in these bacterial infections. Despite the increased focus on this topic, similar regional studies from Southern Europe about AMR and antibiotic administration are lacking. Indeed, the majority of the veterinary literature comes from Northern Europe, Oceania and North America [31,32,33,34,35,36,37], and few recent manuscripts from countries such as Italy or Spain [18,22,38,39] are focused only on AMR and MDR rates. Our study combined information about AMR and antibiotic administration evaluating also their trend over time with the application of the ISCAID guidelines. The knowledge of regional bacterial variations is an essential step to better manage infections, adapting empirical treatments to local AMR results [32]. We investigated the in vitro AMR by using the Kirby–Bauer disc diffusion method, still considered an accurate and reproducible method for AST, in line with the EUCAST guidelines [40].

Specimens from dogs were consistently more frequent, but this could be due to the larger number of dogs attending at the VTH. Nevertheless, the literature on this field reports that dogs, specifically females, tend to have a higher frequency of UTIs compared with cats [8]; on the other hand, cats are described having a higher prevalence of idiopathic cystitis without concurrent bacterial involvement [7,41]. The median age at the moment of sampling (10 years) was in line with previous reports [4,7,42].

In both dogs and cats, the most common bacterial species was *E. coli*, representing around half of the total isolates. This finding is in accordance with previous large studies in Europe, in which the *E. coli* prevalence in UTIs from dogs and cats varies from 34.5% to 59.8% [7,18,19,32,39,43,44,45]. Similarly to humans, its high prevalence as urinary pathogen in pets is mainly due to its plasticity and capacity to share and acquire antimicrobial resistance and virulence genes, which help its persistence [4]. Other frequently isolated species in our study were staphylococci members of the *Staphylococcus intermedius* group (SIG), *K. pneumoniae*, *P. mirabilis*, *E. faecalis* and *S. canis.* Specifically, *S. canis* was associated with a higher frequency in dogs, in line with similar studies from Europe [4,43,46]. On the other hand, bacterial species such as *E. faecalis*, *C. urealyticum*, *S. aureus* and *S. felis* were significantly more prevalent in cats. This differences are supported by other authors [4,43,46,47,48,49] and could reflect a species-specific propensity for some bacterial species causing UTIs. Notably, we did not observe any statistically significant difference about the distribution of species members of the SIG, unlike other studies, which described a lower incidence in cats, mainly due to the absence of *S. pseudintermedius* (the main SIG species in companion animals) in their skin microbiota [50,51].

Regarding the type of specimens, most of them were urine samples (98.8%), and bacterial growth in monoculture was observed in 91.3% of samples. The frequency of mixed infections (8.7%) was lower compared with other reports, describing a frequency from 15.8% to 28.9% [4,5,47]; on the other hand, our results are in line with a large European multicenter study by Marques et al. [7], in which the reported frequency was 5.36%, and also another Italian report by Rampacci et al. [19], in which the frequency was 5% and 6.3% in dogs and cats, respectively. In this latter manuscript, animals were previously treated with antibiotics. In our study as well, a significant proportion of patients—approximately 40%—had received prior antibiotic treatment, which may have contributed to the selection of single-species growth in the specimens.

The application of the ISCAID guidelines to categorize UTIs, highlights that almost half of the specimens came from cases classified as RBC or uUTI. The high proportion of these types of UTIs reflects the fact that our VTH is a tertiary-care facility, including a substantial number of referred, second-opinion patients with complicated infections. When compared with a previous study from the USA [12] that analyzed antimicrobial prescriptions in dogs with suspected UTIs from 2016 to 2018, and with a study conducted in Germany [33] on urinary isolates from cats in a 7-year period, we recorded a considerably higher proportion of RBC and uUTI cases. This discrepancy can be related to the different methodologies used, but also to the geographical and structural variability. Our high proportion of complicated cases is linked with the percentage of specimens (almost 40% in both dogs and cats) taken from patients previously treated with antibiotics, especially in cases classified as CAUTI (66.7%), RBC (65.9%) and uUTI (42.5%). On the other hand, antimicrobial treatment at the time of sampling involved 13.7% of the specimens, indicating that the empirical treatment (antibiotic started before the results from the AST based on clinical evaluation, laboratory and diagnostic imaging data) was infrequent, especially for cases classified as SUB (5.3%) and SBC (10%).

In contrast with the majority of the studies on this topic, focusing on the analysis of quantitative consumption or antibiotic prescriptions [6,52], in our study we analyzed the antibiotic use only in patients with positive specimens; consequently, patients who did not have positive bacteriological samples but still received antibiotic treatment were excluded from the analysis. This could have led to an underestimation of the overall antibiotic consumption for UTIs; nevertheless, data can still be considered a reflection of the local habits of antibiotics prescription. Our results showed that amoxicillin-clavulanate was, not surprisingly, the most used antibiotic. In fact, it is considered a first-line antibiotic for SBC [6,16,53,54,55,56]. Compared with non-potentiated penicillins, amoxicillin-clavulanic acid has similar effects on antimicrobial resistance, although it could negatively impact on the beneficial microbiota [57]. Nevertheless, its efficacy for uUTI (e.g., pyelonephritis) is considered controversial due to the inability to reach adequate therapeutic concentration in renal tissue [16]. A Finnish study by Rantala et al. [58] described trimethoprim-sulfamethoxazole as the most commonly used drug to treat acute UTIs (52%). This antibiotic is considered by the ISCAID a first-line option for UTIs [16], but its use in our study was not reported. The reason behind this discrepancy could be mainly due to the possible adverse effects when used for long periods, such as immune-mediated reactions [16,59] and the absence of a specific veterinary oral formulation in Italy for cats and dogs. Fluoroquinolones, especially marbofloxacin, were the second most used antibiotic class, with higher proportions compared to other studies from the United States [6], but similar to other manuscripts from Denmark, United States and New Zealand [56,60,61]. A Swiss study from Schmitt et al. [62] on cats showed that fluoroquinolones prescriptions for lower UTIs were less frequent in university hospitals compared to private practices. On the other hand, they are still considered a first-line drug to treat pyelonephritis and prostatitis [16], the frequency of which tends to be higher in specialized facilities such as VTHs. In our case, marbofloxacin was more frequently used than enrofloxacin, whose use is reported to have side effects in cats [16]. In cats, several studies also highlight the frequent use of third-generation cephalosporins, such as cefovecin, to treat SBC [13,62,63], mainly for its single-injection use and, therefore, its use as an easier option in animals difficult to medicate such as cats [64]. In our study, only a small proportion of the specimens from cats was treated with third-generation cephalosporins; this finding agrees with the ISCAID guidelines, on which these drugs are not recommended for routine use [16]: also, our data could be partially explained by the fact that in VTHs there are fewer difficulties to administer antibiotics. Notably, in our study, we also reported the sporadic use of piperacillin-tazobactam (with higher proportions in cats), an ureidopenicillin with beta-lactamases inhibitor belonging to the antibiotic class “A” (“Avoid”) from the European Medicine Agency (EMA) [65]. Its use was allowed for critical patients in veterinary hospitals until the official prohibition with the 2022/1255 EU regulation, which entered into force in March 2023.

The non-susceptibility rates highlighted in this study can be considered only a partial reflection of the local epidemiological scenario, mainly influenced by the selective pressure exerted by antimicrobial administration [66]. Indeed, facilities such as the VTH of this study are referral centers with a higher frequency of patients with comorbidities, previously treated with antimicrobials, who need to be checked periodically and with the same pathogen isolated multiple times. Despite perfect duplicates not being included in the study, this could have led to an overassessment of resistance rates. Additionally, the choice to consider intermediate isolates as resistant ones could have been another source of overestimation, given that many antimicrobials considered (such as cephalosporins) can be highly concentrated in urine. In our study, *E. coli* was not surprisingly the most frequent MDR species (*n* = 120), given that it was also the most common. Notably, *E. faecium* showed the highest MDR proportion (80% and 74.6%, respectively), a result that aligns with the previous literature on companion animals [67,68]. In 2016, Marques et al. [7] described that frequencies in uropathogenic bacteria resistance collected in Southern Europe countries were considerably higher compared with Northern Europe: in Italy, 28.99% of *E. coli* isolates were considered MDR, similarly to our results, in which the MDR proportion of *E. coli* was 31.1%. This result took into consideration isolates collected between 2008 and 2013; given the temporal distance with our results, and the speed of AMR spread, comparisons should be made with caution. In our study, the overall non-susceptibility percentage recorded for amoxicillin-clavulanate (21.1%) was relatively low if compared with the other drugs tested, confirming its use as a first-line drug for SBC [16]. Although its use was not reported in our analysis, trimethoprim-sulfamethoxazole showed a similar non-susceptibility percentage (23.4%), suggesting that this antibiotic could be considered as an alternative option for short-term treatment, in order to avoid side effects [69]. On the other hand, non-potentiated penicillin such as amoxicillin, reported to be another first-line drug for SBC by the ISCAID guidelines [16], showed a non-susceptibility percentage higher than 50% especially in dogs, which is the reason why it should be used with caution in these kinds of patients. Such discrepancy between amoxicillin and amoxicillin-clavulanate percentages is similar to a study by KuKanich et al. [70] reporting susceptibility rates of 53% and 92%, respectively. Non-susceptibility towards third-generation cephalosporins was reported to be significantly higher in cats. Again, this could be due to a major tendency to use cefovecin in this species for ease of use, or also to the differences between population size. The percentage of non-susceptibility recorded for enrofloxacin (45.6%) is alarming, but in line with other similar studies from Italy [7,19,46]; this result underlines the need to limit fluoroquinolone use in small animal practice, or in any case its reasoned use in selected patients. Indeed, although they have been classified as Highest Priority Critically Important Antimicrobials (HPCIAs) by the WHO [17], in Italy, they are still frequently used in first-opinion practices due to their broad-spectrum effect and ease of administration [71,72,73,74].

The temporal analysis of the non-susceptibility rates reveals a general decreasing trend over the five semesters for most of the tested drugs; also, the overall MDR rate shows a similar trend. Our result is in line with data obtained in a 2021 study from Thailand by Amphaiphan et al. [75] regarding amoxicillin-clavulanate resistance rates, in which a decrease MDR trend over a 3-year period (from 50% to 15%) was highlighted in uropathogenic bacteria isolated from cats. Our results go together with the decreasing trend of specimens from patients with previous antibiotic treatment, including important ones such as marbofloxacin and piperacillin-tazobactam (whose use was completely stopped at the end of the last semester of observation). Furthermore, our results highlighted a decreasing trend of MDR for specimens from patients under antibiotic treatment at the time of sampling, suggesting a more prudent use as empirical antibiotic therapy. Applying ISCAID guidelines, especially in patients already on antibiotic therapy at the time of inclusion or with previous antibiotic therapy, using targeted molecules and reported timing of therapy, may have decreased the trend of antibiotic resistance selecting non-MDR bacterial populations. Moreover, the significant decrease over time in the proportion of MDR *E. coli* is worth mentioning for two reasons. First, as previously stated, this species was the most commonly found, and also there is evidence that uropathogenic *E. coli* can be shared between humans and pets [76,77,78], so a reduction in MDR over time is important also from a One Health perspective. Additionally, the results from the multivariate analysis confirmed the strict relationship between the previous use of antibiotics (specifically, fluoroquinolones, amoxicillin-clavulanate, piperacillin-tazobactam and third-generation cephalosporins) and multidrug-resistance. Although a clear cause–effect relationship cannot be demonstrated and the timeframe was relatively short, these results are probably linked with the application of the ISCAID guidelines for managing UTIs and should be considered regarding a correct antimicrobial stewardship to reduce the spread of antibiotic resistance. These guidelines provide an evidence-based approach about the diagnosis and management of UTIs and help veterinarians in the decision-making process. Antibiotic use is not only the major driving force for the onset of AMR, but also the most manageable action, so a proper intervention in this field can reduce the related risks. Additionally, this study was conducted in Italy, one of the European countries with the most concerning AMR situations in both human and veterinary medicine [79,80]. Given this geographical context, this significant decreasing trend in both MDR rates and antibiotic administration gives further importance to our results.

This study has some limitations. First, the time lapse of the observation period (30 months) was relatively short for a comprehensive evaluation of the MDR temporal trends. Second, the use of laboratory data could not reflect the overall overview over antibiotic consumption and resistances in our VUH, since uncomplicated cases (for which urine culture was not performed) may have been overlooked [81,82]. However, we anecdotally report that the empirical treatment of UTIs, even in SBC, is a very rare event at our VUH, so our findings could be quite representative of the internal situation. Third, because not all cases were followed directly by the clinicians in the nephrology and urology unit who authored this study, we cannot be sure that the ISCAID guidelines were always applied accurately; however, we consider this unlikely given the percentages reported above.

## 5. Conclusions

This study highlights the impact of AMR in UTIs from companion animals in an Italian VUH, and designed how it could be reduced over time. Our results add novelty into the Italian epidemiological scene and reinforce the role of clinicians through a judicious and rational UTIs management with antibiotics administration oriented by scientific evidence-based guidelines, linked to the local situation. Additionally, our work underlines how a well-defined monitoring system can help in detecting correlations between antibiotics consumption and resistance rates, prioritizing antibiotics use following antimicrobial stewardship. Given that pets are a potential reservoir of resistance towards several antibiotics and are in contact with people, such findings also represent an important challenge from a One Health point of view, especially regarding HCPIAs such as fluoroquinolones and third-generation cephalosporins.

## Figures and Tables

**Figure 1 animals-15-01547-f001:**
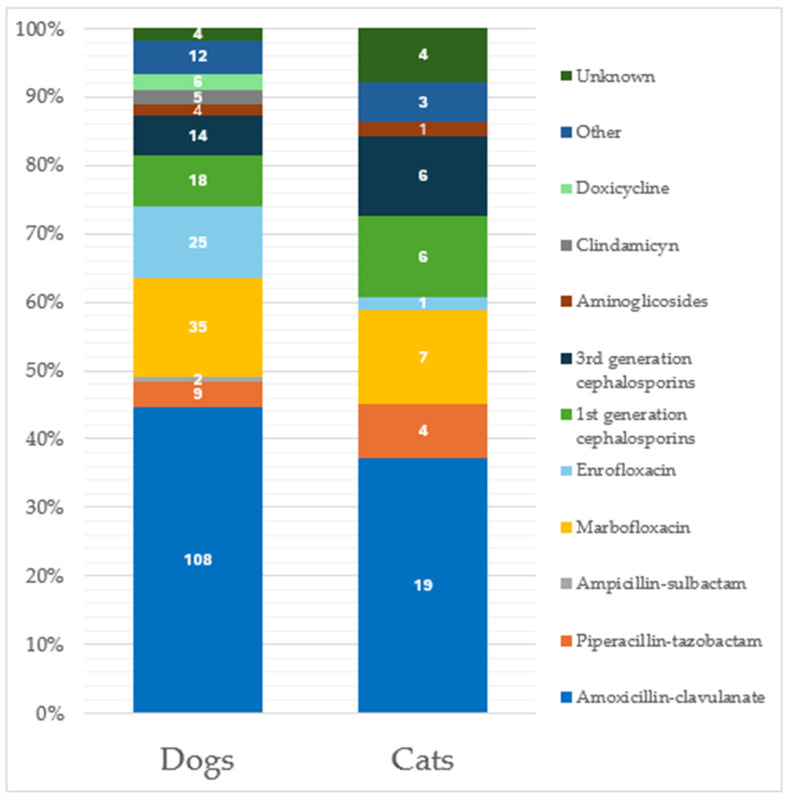
Distribution of antimicrobial administration considering the 265 specimens from patients treated in the previous 90 days. Classification is made considering separately the two species (219 from dogs and 46 from cats). Data labels are shown within the bars.

**Figure 2 animals-15-01547-f002:**
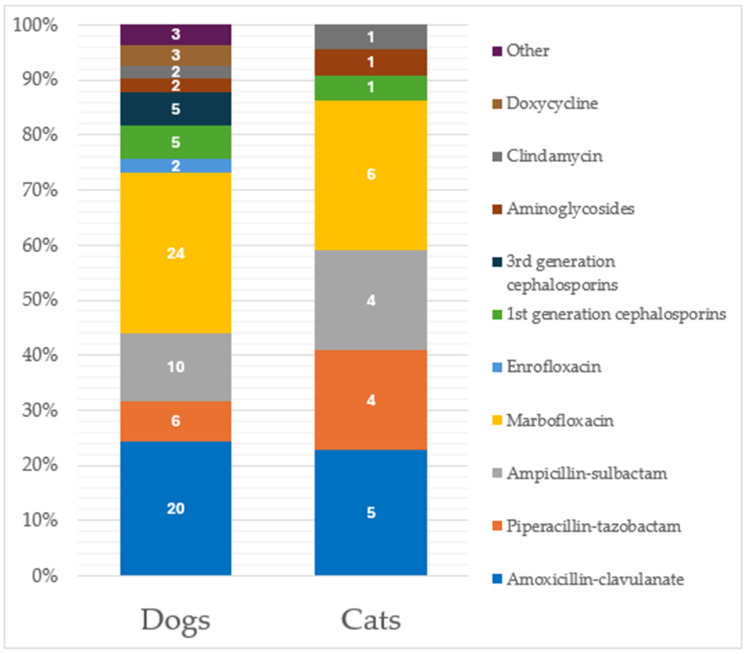
Distribution of antimicrobial administration considering the 92 specimens from patients under antibiotic treatment at the time of sampling. Classification is made considering separately the two species (73 from dogs and 19 from cats). Data labels are shown within the bars.

**Figure 3 animals-15-01547-f003:**
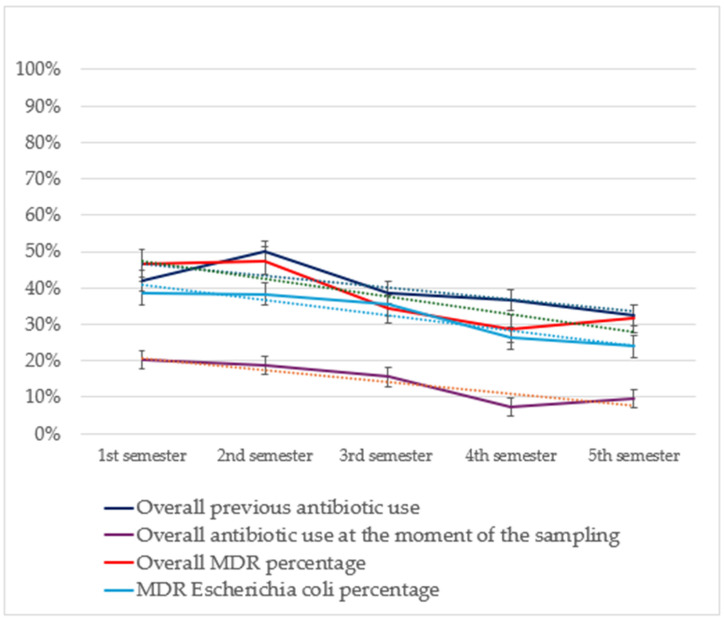
Results from the temporal trend analysis of MDR percentages of the isolates included in the study (in orange) and of the MDR percentage of *E. coli* (in light blue) relating it with antibiotic use both in the previous 90 days (in dark blue) and at the time of sampling (in violet). Tendency lines (dashed) and 95% confidence intervals are also shown. MDR, multi-drug resistance bacteria.

**Table 1 animals-15-01547-t001:** Distribution of the bacterial species isolated from positive specimens included in the study, considering both dogs and cats. *p* values comparing bacterial species between canine and feline isolates are shown, and values considered statistically significant are shown in bold.

Bacterial Species	Dogs (*n*)	Dogs (%)	Cats (*n*)	Cats (%)	*p* Value
Total	602		127		
*Citrobacter* spp.	6	1.0%	0	0.0%	0.590
Coagulase-negative *Staphylococci* (CoNS) other than *Staphylococcus felis*	4	0.7%	5	3.9%	**0.010**
*Corynebacterium urealyticum*	2	0.3%	3	2.4%	**0.039**
*Enterobacter cloacae*	15	2.5%	1	0.8%	0.330
*Enterococcus faecalis*	33	5.5%	19	15.0%	**<0.001**
*Enterococcus faecium*	12	2.0%	3	2.4%	0.733
*Escherichia coli*	318	52.8%	58	45.7%	0.171
*Klebsiella pneumoniae*	39	6.5%	6	4.7%	0.547
Other *Enterobacter* spp. ^1^	5	0.8%	1	0.8%	1
Other *Enterococcus* spp. ^2^	2	0.3%	1	0.8%	0.437
Other gram-negatives ^3^	11	1.8%	0	0.0%	0.227
Other *Streptococcus* spp.	5	0.8%	0	0.0%	0.593
*Proteus mirabilis*	33	5.5%	7	5.5%	1
*Pseudomonas aeruginosa*	22	3.6%	3	2.4%	0.599
*Staphylococcus aureus*	2	0.3%	3	2.4%	**0.039**
*Staphylococcus felis*	0	0.0%	7	5.5%	**<0.001**
*Staphylococcus intermedius* group (SIG)	58	9.6%	9	7.1%	0.403
*Streptococcus canis*	35	5.8%	1	0.8%	**0.012**

*n*, number of specimens; ^1^*Enterobacter hormaechei* (*n* = 3), *Enterobacter ludwingii* (*n* = 2), *Enterobacter bugandensis* (*n* = 1), ^2^
*Streptococcus gallolyticus* (*n* = 3), *Streptococcus agalactyae* (*n* = 1), *Streptococcus dysgalactiae* (*n* = 1); ^3^ Species identified were *Burkholderlia cepacia* (*n* = 1), *Canicola haemoglobinophilus* (*n* = 1), *Klebsiella aerogenes* (*n* = 3), *Klebsiella oxytoca* (*n* = 3), *Klebsiella variicola* (*n* = 1), *Moraxella osloensis* (n = 1) and *Stenotrophomonas maltophilia* (*n* = 1).

**Table 2 animals-15-01547-t002:** Descriptive statistics of patients’ hospitalization data considering all the 670 specimens collected from dogs and cats.

Hospitalization Data	Total Specimens (*n* = 670)	Total Specimens (%)	Specimens from Cats (*n* = 116)	Specimens from Cats (%)	Specimens from Dogs (*n* = 554)	Specimens from Dogs (%)
Previous hospitalization/surgery in the past 30 days	119	17.8%	27	23.3%	92	16.6%
Hospitalization at the time of sampling	167	24.9%	40	34.5%	127	22.9%
Hospitalization in intensive care unit	79	11.8%	18	15.5%	61	11%
Surgery at the time of sampling	47	7%	10	8.6%	37	6.7%

*n*, number of specimens.

**Table 3 animals-15-01547-t003:** Distribution of the 670 specimens according to the ISCAID classification, considering animal species (dogs/cats), single or mixed infection and the use of antibiotics (previously or at the time of sampling).

ISCAID Classification	Total Cases (%)	Cases from Dogs *n* (%)	Cases from Cats *n* (%)	Single-Species Infection *n* (%)	Mixed Infection *n* (%)	Cases with Previous Antimicrobial Use *n* (%)	Cases with Antimicrobial Use at the Time of Sampling *n* (%)
SUB	114 (17%)	99 (17.9%)	15 (12.9%)	103 (90.3%)	11 (9.7%)	30 (26.3%)	6 (5.3%)
SBC	190 (28.4%)	159 (28.7%)	31 (26.7%)	178 (93.7%)	12 (6.3%)	38 (20%)	19 (10%)
RBC	166 (24.8%)	132 (23.7%)	34 (29.3%)	143 (86.2%)	23 (13.8%)	110 (65.9%)	20 (12%)
uUTI	167 (24.9%)	137 (24.8%)	30 (25.9%)	155 (92.8%)	12 (7.2%)	71 (42.5%)	38 (22.8%)
CAUTI	12 (1.8%)	8 (1.4%)	4 (3.4%)	12 (100%)	0 (0%)	8 (66.7%)	5 (41.7%)
BP	16 (2.4%)	16 (2.8%)	0 (0%)	16 (100%)	0 (0%)	6 (37.5%)	3 (18.8%)
NC	5 (0.7%)	5 (0.9%)	0 (0%)	5 (100%)	0 (0%)	3 (60%)	1 (20%)
Total	670	554	116	612 (91.3%)	58 (8.7%)	266 (39.7%)	92 (13.7%)

BP, Bacterial prostatitis; CAUTI, catheter-associated urinary tract infection; ISCAID, International Society for Companion Animals Infectious Disease; *n*, Number of specimens; NC, not classifiable; RBC, recurrent bacterial cystitis; SBC, sporadic bacterial cystitis; SUB, subclinical bacteriuria; uUTI, upper urinary tract infection.

**Table 4 animals-15-01547-t004:** Non-susceptibility percentages of the bacterial isolates included in the study for each tested antibiotic. *p* values comparing non-susceptibility percentages between canine and feline isolates are shown, and the ones considered statistically significant are in bold.

Antimicrobial Tested	Non-Susceptibility % Among all Bacterial Population (*n* = 729)	Non-Susceptibility % of Bacterial Isolates from Dogs (*n* = 602)	Non-Susceptibility % of Bacterial Isolates from Cats (*n* = 127)	*p* Value
Amikacin	10.6% (65/612)	10.1% (52/512)	13% (13/100)	0.398
Gentamicin	17.3% (119/687)	16.6% (93/561)	20.6% (26/126)	0.277
Ampicillin	52.6% (328/623)	54.5% (276/506)	44.4% (52/117)	**0.048**
Amoxicillin-clavulanate	23.2% (156/671)	22.1% (122/552)	28.5% (34/119)	0.129
Piperacillin-tazobactam	17% (123/722)	16.2% (97/598)	21% (26/124)	0.201
Cephazolin/Cephalothin	28.8% (173/601)	27.5% (139/505)	35.4% (34/96)	0.177
Ceftiofur	23.1% (145/627)	21.3% (113/529)	32.6% (32/102)	**0.028**
Tetracycline	41.6% (286/688)	40.8% (232/568)	45% (54/120)	0.401
Erythromycin	58.1% (75/129)	61.5% (64/104)	44% (11/25)	0.110
Clindamycin	62.7% (84/134)	65.1% (69/106)	53.5% (15/29)	0.188
Enrofloxacin	45.2% (327/729)	44.4% (265/599)	50% (62/124)	0.241
Trimethoprim- sulfamethoxazole	22% (138/628)	21.3% (113/530)	25.5% (25/98)	0.357

*n*, number of specimens.

**Table 5 animals-15-01547-t005:** Distribution of the multidrug-resistant (MDR) isolates per bacterial species. The percentage of MDR isolates over the total number of MDR isolates (*n* = 272) is shown, as well as the proportion of MDR isolates to total number of isolates of the same species.

Bacterial Species	*n* of MDR Isolates	% of MDR Isolates	% of MDR Isolates Within the Species
*E. coli*	120	44.1%	31.9% (120/376)
*Enterobacter* spp.	7	2.6%	31.8% (7/22)
*E. faecium*	12	4.4%	80.0% (12/15)
*E. faecalis*	9	3.3%	17.3% (9/52)
*C. urealyticum*	1	0.4%	20.0% (1/5)
*K. pneumoniae*	25	9.2%	55.6% (25/45)
*P. aeruginosa*	7	2.6%	28.0% (7/25)
*P. mirabilis*	8	2.9%	20.0% (8/40)
*Staphylococcus intermedius* group (SIG)	50	18.4%	74.6% (50/67)
*S. canis*	21	7.7%	58.3% (21/36)
*S. aureus*	3	1.1%	60.0% (1/5)
*K. oxytoca*	1	0.4%	33.3% (1/3)
Coagulase-negative *Staphylococci* (CoNS)	5	1.8%	31.3% (5/16)
Other *Streptococcus* spp.	3	1.1%	60% (3/5)

*n*, number of specimens; MDR: multidrug-resistant.

## Data Availability

Data are available in the database of the Veterinary university Hospital.

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
