# Peer review of "Antimicrobial Resistance in Companion Animals: A 30-Month Analysis on Clinical Isolates from Urinary Tract Infections in a Veterinary Hospital"

_animals, 2025, doi:10.3390/ani15111547_

Round 1
Reviewer 1 Report
Comments and Suggestions for Authors
Summary: First of all, I would like to congratulate the Authors for their excellent work. This manuscript presents a comprehensive and relevant evaluation of antimicrobial resistance patterns in bacterial urinary tract infections among companion animals, highlighting the impact of applying ISCAID guidelines in clinical practice. The manuscript is well-written, logically structured, and easy to follow. However, some points require minor clarification or revision to enhance its quality further.
Specific Comments:
Line 37: Please ensure that the "n" in the expression "Multidrug resistance (MDR) percentage was 37.3% (n=272)" is italicised, as this is the correct formatting for statistical and numerical reporting. It should appear as "(n=272)". Apply this correction throughout the manuscript.
Line 99: The list "urine, bladder stones, urinary bladder biopsies" should include "and" before the last item to improve readability. Please revise as: "urine, bladder stones, and urinary bladder biopsies".
Line 151: The phrase "Odds ratio (OR) and 95% CI were calculated" mentions "95% CI" again, which was previously stated at line 144. Consider rephrasing to avoid redundancy—for instance, "Odds ratios (OR) were also calculated."
Line 152 and Line 157: The sentence "Normality and heteroskedasticity of data were assessed with Shapiro–Wilk test and Levene’s test." appears duplicated. Please remove one of these repetitions to improve readability.
Line 159: Please correct "MedCalc software version v22.009" to "MedCalc® statistical software version 22.009 (MedCalc Software Ltd, Ostend, Belgium)" for accuracy and proper software citation.
Line 166: Consider presenting the age data clearly and consistently as follows:
"Age distribution of sampled patients ranged from 1 year (≤ 1 year) to 17 years in dogs and from 1 year (≤ 1 year) to 19 years in cats, with a median age of 10 years for both species (interquartile range: X to Y years)."
If possible, include the interquartile range to enhance statistical clarity.
Line 175: Table 1: The letter "n" indicating the number of cases should be italicised consistently throughout the table for proper statistical formatting (e.g., n). Please revise accordingly.
Line 176: In Table 1, the p-value for Citrobacter spp. should be presented with three decimal places for consistency with the rest of the table. Please revise it to read "0.590".
Line 176: In Table 1, "Streptococcus canis" should be italicised as it is a bacterial species name. Please correct it accordingly.
Line 176: In the legend of Table 1, "N" should be changed to lowercase and italicised ("n") to indicate the number of specimens correctly. The corrected version should read:
"CONS, coagulase-negative staphylococci; n, number of specimens; SIG, Staphylococcus intermedius group. 1Species identified were Burkholderia cepacia (n=1), Canicola haemoglobinophilus (n=1), Klebsiella aerogenes (n=3), Klebsiella oxytoca (n=3), Klebsiella variicola (n=1), Moraxella osloensis (n=1), and Stenotrophomonas maltophilia (n=1)."
Line 189: There is an inconsistency between the text and Table 2 regarding the total number of specimens. The text and Table 2 report 670 specimens, whereas the Table 2 title mistakenly indicates 671 specimens. Please correct this discrepancy to ensure consistency throughout the manuscript. Also correct n for statistical and numerical reporting.
Line 200: Please revise the legend of Figure 1 to improve clarity:
Figure 1. Distribution of antimicrobial administration in patients treated in the previous 90 days. Data are presented separately for each species (dogs and cats).
Line 207: Please revise the legend of Figure 2 to improve clarity:
Figure 2. (…) Data are presented separately for each species (dogs and cats).
Line 209: Please consider revise the sentence to improve readability and clarity as follows:
"In cats, similar percentages were observed, with 39.7% (n=46) of specimens obtained from animals previously treated with antimicrobials; specifically, 39 (84.8%) were treated with one antibiotic, 6 (13.0%) with two antibiotics, and 1 (2.2%) with three antibiotics."
Line 224: Table 3: Please correct the total number of specimens from '671 specimens' to '670 specimens' to ensure consistency with the rest of the manuscript. Also correct n for statistical and numerical reporting.
Line 242:
- Please verify the total number of isolates indicated (n=742) for accuracy, as it does not match the previously stated total (n=729).
- Correct the numbers of isolates presented for dogs (n=613) and cats (n=129) to match previously reported values (dogs: 602, cats: 127).
- Replace uppercase 'N' with lowercase italicised 'n' in the legend: 'n, number of specimens'.
- Correct antimicrobial names: 'Erythromicin' to 'Erythromycin', 'Clyndamicin' to 'Clindamycin', and 'Cephazolin/Cephalotin' to 'Cefazolin/Cephalothin'.
- Ensure consistent formatting of percentages and isolate counts throughout the table.
Line 253: When using Odds Ratios (OR), the standard population or reference category must be clearly stated in the methods section. Please clarify explicitly which reference population or baseline category was used for the temporal trend OR analysis to improve methodological transparency.
Line 257: In the confidence interval "CI 0,655 to 0,874", please replace the commas with decimal points to comply with standard scientific English formatting. It should read: "CI 0.655 to 0.874".
Line 297-298: It would be beneficial to elaborate slightly more on why E. coli is consistently the most common isolate in companion animal UTIs.
Line 319: Please consider revise the sentence to improve readability and clarity as follows:
"In our study as well, a significant proportion of patients – approximately 40% – had received prior antibiotic treatment, which may have contributed to the selection of single-species growth in the specimens."
Lines 323-329: Clarify briefly why mixed infections might occur less frequently in this study compared to other European studies.
Line 326: Please correct the phrase “When comparing with a previous study from the US” to: “When compared with a previous study from the USA...”
This improves grammatical structure and ensures formal consistency with scientific writing conventions.
Lines 346-355: Consider briefly discussing why trimethoprim-sulfamethoxazole use was notably absent, despite ISCAID guidelines recommending it as first-line therapy.
Lines 387-390: Given the importance of MDR E. coli, consider highlighting any significant trends observed specifically for this organism throughout the study.
Line 408: Please spell out the acronym HPCIAs at its first occurrence in the text for clarity. It should read: “Highest Priority Critically Important Antimicrobials (HPCIAs)”.
Ensure the acronym is used consistently thereafter.
Lines 410-420: Clearly linking the decline in MDR to the ISCAID guidelines is compelling. Consider briefly mentioning any specific measures from these guidelines that were particularly impactful.
Lines 425-434: Address whether the trend analysis considered other external factors (e.g., regulatory changes, local public health campaigns) that might have influenced antimicrobial prescribing practices.
Line 453: As the full term Highest Priority Critically Important Antimicrobials was already introduced earlier (line 408), please use the acronym HPCIAs in line 453 to avoid repetition and maintain consistency.
Line 477: Antimicrobial names in Table A2:
Please correct the following names to their standard forms: “Erythromicin” to “Erythromycin”, “Cephazolin/cephalotin” to “Cefazolin/Cephalothin”.
References: Please revise the reference list according to the following:
- Reference 3: The page range appears incorrect: “325–325”. Please verify the correct page numbers and amend accordingly.
- Reference 10: There is a typographical error in the journal title: “Veterinary Internal Medicne” should be corrected to “Veterinary Internal Medicine”.
- Reference 18: Please correct the word “Perfomance” to “Performance” in the title.
- Reference 36: The title capitalisation is inconsistent with the rest of the reference list. Consider adjusting to sentence case or the chosen journal style (e.g., only the first word and proper nouns capitalised).
- Reference 19: The authors of the cited book chapter are missing. Please include the names of the authors before the editors of the book.
- Reference 20: This reference is in Italian. If it is an official technical document, please indicate that (e.g., “Technical Report”) and consider providing an English translation of the title in square brackets for clarity.
- Reference Duplication: References 23 and 37 appear to be duplicated.
Author Response
Comments 1:
Line 37: Please ensure that the "n" in the expression "Multidrug resistance (MDR) percentage was 37.3% (n=272)" is italicized, as this is the correct formatting for statistical and numerical reporting. It should appear as "(n=272)". Apply this correction throughout the manuscript.
Response 1: Thank you for pointing this out. We have corrected the format as requested throughout the manuscript.
Comments 2:
Line 99: The list "urine, bladder stones, urinary bladder biopsies" should include "and" before the last item to improve readability. Please revise as: "urine, bladder stones, and urinary bladder biopsies".
Response 2: The sentence was corrected as requested.
Comments 3:
Line 151: The phrase "Odds ratio (OR) and 95% CI were calculated" mentions "95% CI" again, which was previously stated at line 144. Consider rephrasing to avoid redundancy—for instance, "Odds ratios (OR) were also calculated."
Response 3: The sentence was corrected as requested.
Comments 4:
Line 152 and Line 157: The sentence "Normality and heteroskedasticity of data were assessed with Shapiro–Wilk test and Levene’s test." appears duplicated. Please remove one of these repetitions to improve readability.
Response 4: We have removed the repetition as requested.
Comments 5:
Line 159: Please correct "MedCalc software version v22.009" to "MedCalc® statistical software version 22.009 (MedCalc Software Ltd, Ostend, Belgium)" for accuracy and proper software citation.
Response 5: We have corrected the sentence for a proper citation, as requested.
Comments 6:
Line 166: Consider presenting the age data clearly and consistently as follows:
"Age distribution of sampled patients ranged from 1 year (≤ 1 year) to 17 years in dogs and from 1 year (≤ 1 year) to 19 years in cats, with a median age of 10 years for both species (interquartile range: X to Y years)."
If possible, include the interquartile range to enhance statistical clarity.
Response 6: We have corrected the data presentation as requested.
Comments 7:
Line 175: Table 1: The letter "n" indicating the number of cases should be italicised consistently throughout the table for proper statistical formatting (e.g., n). Please revise accordingly.
Response 7: We have revised it, thanks for the suggestion
Comments 8:
Line 176: In Table 1, the p-value for Citrobacter spp. should be presented with three decimal places for consistency with the rest of the table. Please revise it to read "0.590".
Response 8: We have modified the p-value as requested.
Comments 9:
Line 176: In Table 1, "Streptococcus canis" should be italicised as it is a bacterial species name. Please correct it accordingly.
Response 9: We have modified the term as requested.
Comments 10:
Line 176: In the legend of Table 1, "N" should be changed to lowercase and italicised ("n") to indicate the number of specimens correctly. The corrected version should read:
"CONS, coagulase-negative staphylococci; n, number of specimens; SIG, Staphylococcus intermedius group. 1Species identified were Burkholderia cepacia (n=1), Canicola haemoglobinophilus (n=1), Klebsiella aerogenes (n=3), Klebsiella oxytoca (n=3), Klebsiella variicola (n=1), Moraxella osloensis (n=1), and Stenotrophomonas maltophilia (n=1)."
Response 10: We have modified as requested.
Comments 11:
Line 189: There is an inconsistency between the text and Table 2 regarding the total number of specimens. The text and Table 2 report 670 specimens, whereas the Table 2 title mistakenly indicates 671 specimens. Please correct this discrepancy to ensure consistency throughout the manuscript. Also correct n for statistical and numerical reporting.
Response 11: The reviewer is right, the text of the table was modified.
Comments 12:
Line 200: Please revise the legend of Figure 1 to improve clarity:
Figure 1. Distribution of antimicrobial administration in patients treated in the previous 90 days. Data are presented separately for each species (dogs and cats).
Response 12: We have modified the sentence for greater clarity, as requested.
Comments 13:
Line 207: Please revise the legend of Figure 2 to improve clarity:
Figure 2. (…) Data are presented separately for each species (dogs and cats).
Response 13: The sentence was modified as requested.
Comments 14:
Line 209: Please consider revise the sentence to improve readability and clarity as follows:
"In cats, similar percentages were observed, with 39.7% (n=46) of specimens obtained from animals previously treated with antimicrobials; specifically, 39 (84.8%) were treated with one antibiotic, 6 (13.0%) with two antibiotics, and 1 (2.2%) with three antibiotics."
Response 14: the reviewer is right, and the sentence was modified for greater clarity according with the suggestions.
Comments 15:
Line 224: Table 3: Please correct the total number of specimens from '671 specimens' to '670 specimens' to ensure consistency with the rest of the manuscript. Also correct n for statistical and numerical reporting.
Response 15: We have corrected the data as requested.
Comments 16:
Line 242:
- Please verify the total number of isolates indicated (n=742) for accuracy, as it does not match the previously stated total (n=729).
- Correct the numbers of isolates presented for dogs (n=613) and cats (n=129) to match previously reported values (dogs: 602, cats: 127).
- Replace uppercase 'N' with lowercase italicised 'n' in the legend: 'n, number of specimens'.
- Correct antimicrobial names: 'Erythromicin' to 'Erythromycin', 'Clyndamicin' to 'Clindamycin', and 'Cephazolin/Cephalotin' to 'Cefazolin/Cephalothin'.
- Ensure consistent formatting of percentages and isolate counts throughout the table.
Response 16: the reviewer is right, and the text of the table was modified following the suggestions.
Comments 17:
Line 253: When using Odds Ratios (OR), the standard population or reference category must be clearly stated in the methods section. Please clarify explicitly which reference population or baseline category was used for the temporal trend OR analysis to improve methodological transparency.
Response 17: the reviewer is right, and a sentence in the methods section was added.
Comments 18:
Line 257: In the confidence interval "CI 0,655 to 0,874", please replace the commas with decimal points to comply with standard scientific English formatting. It should read: "CI 0.655 to 0.874".
Response 18: We have replaced commas with decimal points as requested.
Comments 19:
Line 297-298: It would be beneficial to elaborate slightly more on why E. coli is consistently the most common isolate in companion animal UTIs.
Response 19: The reviewer is right, and a sentence was added.
Comments 20:
Line 319: Please consider revise the sentence to improve readability and clarity as follows:
"In our study as well, a significant proportion of patients – approximately 40% – had received prior antibiotic treatment, which may have contributed to the selection of single-species growth in the specimens."
Response 20: the sentence was modified following the suggestions.
Comments 21:
Lines 323-329: Clarify briefly why mixed infections might occur less frequently in this study compared to other European studies.
Response 21: As stated in lines 348-357, one of the main reasons for the lower frequence of mixed infections could be due to the high proportion of patients previously treated, that could have contributed to the selection of a single species.
Comments 22:
Line 326: Please correct the phrase “When comparing with a previous study from the US” to: “When compared with a previous study from the USA...”
Response 22: We have corrected the phrase as requested.
Comments 23:
Lines 346-355: Consider briefly discussing why trimethoprim-sulfamethoxazole use was notably absent, despite ISCAID guidelines recommending it as first-line therapy.
Response 23: as stated in lines 388-392, the absence of reported trimethoprim-sulfamethoxazole use could be mainly due to two factors: the possible adverse effects when used for long periods, and the absence of a specific veterinary oral formulation in Italy for cats and dogs.
Comments 24:
Lines 387-390: Given the importance of MDR E. coli, consider highlighting any significant trends observed specifically for this organism throughout the study.
Response 24: The reviewer is right, and the temporal trend of MDR E.coli was added in figure 3.
Comments 25:
Line 408: Please spell out the acronym HPCIAs at its first occurrence in the text for clarity. It should read: “Highest Priority Critically Important Antimicrobials (HPCIAs)”.
Ensure the acronym is used consistently thereafter.
Response 25: We have done it as requested.
Comments 26:
Lines 410-420: Clearly linking the decline in MDR to the ISCAID guidelines is compelling. Consider briefly mentioning any specific measures from these guidelines that were particularly impactful.
Response 26: The sentence was modified to better understand the clinical implications of applying ISCAID guidelines (lines 463-470 and 491-495).
Comments 27:
Lines 425-434: Address whether the trend analysis considered other external factors (e.g., regulatory changes, local public health campaigns) that might have influenced antimicrobial prescribing practices.
Response 27: Considering the trend analysis, other factors such as local public health campaigns, which took into account human antibiotic prescriptions, hardly influenced companion animals’ antibiotic prescriptions, for which veterinary guidelines were applied.
Comments 28:
Line 453: As the full term Highest Priority Critically Important Antimicrobials was already introduced earlier (line 408), please use the acronym HPCIAs in line 453 to avoid repetition and maintain consistency.
Response 28: We have done it as requested.
Comments 29:
Line 477: Antimicrobial names in Table A2:
Please correct the following names to their standard forms: “Erythromicin” to “Erythromycin”, “Cephazolin/cephalotin” to “Cefazolin/Cephalothin”.
References: Please revise the reference list according to the following:
- Reference 3: The page range appears incorrect: “325–325”. Please verify the correct page numbers and amend accordingly.
- Reference 10: There is a typographical error in the journal title: “Veterinary Internal Medicne” should be corrected to “Veterinary Internal Medicine”.
- Reference 18: Please correct the word “Perfomance” to “Performance” in the title.
- Reference 36: The title capitalisation is inconsistent with the rest of the reference list. Consider adjusting to sentence case or the chosen journal style (e.g., only the first word and proper nouns capitalised).
- Reference 19: The authors of the cited book chapter are missing. Please include the names of the authors before the editors of the book.
- Reference 20: This reference is in Italian. If it is an official technical document, please indicate that (e.g., “Technical Report”) and consider providing an English translation of the title in square brackets for clarity.
- Reference Duplication: References 23 and 37 appear to be duplicated.
Response 29: We have corrected as requested.
Reviewer 2 Report
Comments and Suggestions for Authors
Scarpellini et al. conducted a comprehensive study analyzing uropathogens in a European veterinary university hospital, their antimicrobial resistance profiles, and trends following the implementation of ISCAID guidelines for urinary tract infections. The manuscript is generally well written and addresses most relevant aspects. However, further clarification is needed regarding the specific measures taken in accordance with the ISCAID guidelines to show the novelty of this paper in comparison with the paper from 2023 of the same group.
Specific comments:
- Line 20: Fluoroquinolones are no first-line antibiotics. Please revise accordingly.
- Line 60: A reference citing one or more studies on the number of affected animals would be more appropriate here than the previous ISCAID guideline.
- Line 64: The study by Weese et al. (2021), addressing antimicrobial prescription in cats, would be more suitable than citing the current ISCAID guideline.
- Line 72: Please remove the extra space.
- Lines 75–76: Please remove the plural “s” from "UTIs".
- Lines 93–95: Please clarify which exact guidelines were implemented and how (internal training sessions, circulars, etc.). What specifically changed compared to the prior routine? This section lacks detail, which makes later references to a decrease in MDR bacteria hart to follow. Further elaboration would also help other clinics understand and possibly adopt your approach.
- Line 98: Only Table A1 is included in the Appendix.
- Line 123: Please delete "Table A1".
- Lines 124–127: Which CLSI document was used to interpret the veterinary specific breakpoints? If it was not the most recent version (CLSI VET01S ED7:2024), please verify whether any breakpoints have changed and update accordingly.
- Lines 128–130: When referring to “expected phenotypes,” do you mean intrinsic resistance? Please note that the MDR definition by Magiorakos et al. excludes intrinsic resistance. If intrinsic resistance was not tested from the start, it is excluded automatically. Please confirm that this it was considered in your analysis.
- Lines 157–158: This sentence appears identically in lines 152–153. Please remove one of them.
- Line 165: Please include the percentage for the 127 isolates according to the canine data.
- Line 177: Please correct "CoNS" or remove it along with "SIG," as both are explained in the table. If "SIG" is retained, please correct to “Staphylococcus intermedius group”.
- Line 178: Correct “Species” (capitalization).
- Figures 1 & 2: Please correct all spelling mistakes. Not all antimicrobials are listed in the legends. Data labels within the bars are missing. Number of animals are missing in the line 200-201 and 207-208. Additionally in figure 2, the color used for AMC (assumed, as it's not in the legend) differs between dog and cat datasets. Since the legend is incomplete, it is unclear whether this was intentional—especially as line 216–217 states that no significant differences in antimicrobial use were observed.
- Lines 227–230: Capitalization of the first letter varies inconsistently. Please standardize.
- Line 235: Remove double spacing.
- Table 4: Please correct the spelling of “Clindamycin” and “Cefazolin”.
Additionally, the study would be greatly improved by displaying the distribution of MDR isolates per bacterial species, rather than only providing the overall number of MDR isolates.
- Line 301: Please correct “staphylococci” (lowercase "s").
- Line 308: The manuscript does not report the prevalence of SIG group members; please either delete this sentence or present the corresponding data in the results.
- Line 310: Please correct to pseudintermedius, which is the predominant SIG species in the microbiome of companion animals, not animals in general.
- Lines 367–369: Please review this sentence for clarity and consistency.
- Line 373: Remove double spacing.
- Line 398: Amoxicillin, not ampicillin, is a first-line antimicrobial for the treatment of UTIs. Please correct.
- Line 412: Remove double spacing.
- Line 432: Please correct to “European countries”.
- Lines 451–454: Please rephrase the final sentence for clarity and consistency. Also, use “3rd generation” instead of “third generation”.
Reference 16: Please provide the chapter title and page number.
Reference 33: If the reviewer is not mistaken, this appears to be a publication by the same authors, based (at least partially) on the same dataset, published in 2023. Therefore, this reference is missing in the Material and Methods and would be appropriate in line 93, where origin of the samples is first mentioned, as well as in the rest of the methods.
Given the overlap with the previous publication, it is particularly important to clearly explain how the ISCAID guidelines were applied in this study, as this represents the main methodological difference and justifies the current work as a distinct contribution.
Reference 49: This is the wrong reference. Please correct.
Author Response
Comments 1:
Line 20: Fluoroquinolones are no first-line antibiotics. Please revise accordingly.
Response 1: The sentence was modified to better understand the role of first – line drugs of penicillins for sporadic cystitis and fluoroquinolones for pyelonephritis.
Comments 2:
Line 60: A reference citing one or more studies on the number of affected animals would be more appropriate here than the previous ISCAID guideline.
Response 2: We have modified it as requested.
Comments 3:
Line 64: The study by Weese et al. (2021), addressing antimicrobial prescription in cats, would be more suitable than citing the current ISCAID guideline.
Response 3: The reviewer is right, and the reference was modified.
Comments 4:
Line 72: Please remove the extra space
Response 4: We have removed it
Comments 5:
Lines 75–76: Please remove the plural “s” from "UTIs".
Response 5: We have removed it as requested.
Comments 6:
Lines 93–95: Please clarify which exact guidelines were implemented and how (internal training sessions, circulars, etc.). What specifically changed compared to the prior routine? This section lacks detail, which makes later references to a decrease in MDR bacteria hart to follow. Further elaboration would also help other clinics understand and possibly adopt your approach.
Response 6: The sentence was modified as requested.
Comments 7:
Line 98: Only Table A1 is included in the Appendix.
Response 7: The reviewer is right, we have modified the Appendix
Comments 8:
Line 123: Please delete "Table A1".
Response 8: We have deleted it.
Comments 9:
Lines 124–127: Which CLSI document was used to interpret the veterinary specific breakpoints? If it was not the most recent version (CLSI VET01S ED7:2024), please verify whether any breakpoints have changed and update accordingly.
Response 9: The CLSI document initially used to interpret veterinary specific breakpoints was the 2020 version (CLSI VET01S ED5: 2020). Subsequently, they have been updated according to the most recent one (CLSI VET01S ED7:2024). The sentence was corrected.
Comments 10:
Lines 128–130: When referring to “expected phenotypes,” do you mean intrinsic resistance? Please note that the MDR definition by Magiorakos et al. excludes intrinsic resistance. If intrinsic resistance was not tested from the start, it is excluded automatically. Please confirm that this it was considered in your analysis.
Response 10: the reviewer is right, the term “expected phenotypes” refers to intrinsic resistances that were excluded from the subsequent analysis. The sentence was clarified.
Comments 11:
Lines 157–158: This sentence appears identically in lines 152–153. Please remove one of them.
Response 11: We have removed one sentence.
Comments 12:
Line 165: Please include the percentage for the 127 isolates according to the canine data.
Response 12: We have included the percentage as requested.
Comments 13:
Line 177: Please correct "CoNS" or remove it along with "SIG," as both are explained in the table. If "SIG" is retained, please correct to “Staphylococcus intermedius group”.
Response 13: We have modified it as requested.
Comments 14:
Line 178: Correct “Species” (capitalization).
Response 14: We have corrected it.
Comments 15:
Figures 1 & 2: Please correct all spelling mistakes. Not all antimicrobials are listed in the legends. Data labels within the bars are missing. Number of animals are missing in the line 200-201 and 207-208. Additionally in figure 2, the color used for AMC (assumed, as it's not in the legend) differs between dog and cat datasets. Since the legend is incomplete, it is unclear whether this was intentional—especially as line 216–217 states that no significant differences in antimicrobial use were observed.
Response 15: the reviewer is right, and the two figures were corrected following the indications. Additionally, the labels were added and the colors uniformized.
Comments 16:
Lines 227–230: Capitalization of the first letter varies inconsistently. Please standardize.
Response 16: We have standardized it.
Comments 17:
Line 235: Remove double spacing.
Response 17: We have removed it.
Comments 18:
Table 4: Please correct the spelling of “Clindamycin” and “Cefazolin”.
Response 18: We have corrected it.
Comments 19:
Additionally, the study would be greatly improved by displaying the distribution of MDR isolates per bacterial species, rather than only providing the overall number of MDR isolates.
Response 19: the reviewer is right, and a Table (table 5) showing this distribution was added, described (lines 278-80) and discussed (lines 422-425).
Comments 20:
Line 301: Please correct “staphylococci” (lowercase "s").
Response 20: We have corrected it.
Comments 21:
Line 308: The manuscript does not report the prevalence of SIG group members; please either delete this sentence or present the corresponding data in the results.
Response 21: The distribution of SIG group members (9.6% in dogs and 7.1% in cats) is reported in Table 1. The sentence was modified.
Comments 22:
Line 310: Please correct pseudintermedius, which is the predominant SIG species in the microbiome of companion animals, not animals in general.
Response 22: We have corrected it.
Comments 23:
Lines 367–369: Please review this sentence for clarity and consistency.
Line 373: Remove double spacing.
Response 23: We have reviewed both sentences as requested.
Comments 24:
Line 398: Amoxicillin, not ampicillin, is a first-line antimicrobial for the treatment of UTIs. Please correct.
Response 24: the reviewer is right, and the sentence was corrected.
Comments 25:
Line 412: Remove double spacing.
Response 25: We have removed it.
Comments 26:
Line 432: Please correct to “European countries”.
Response 26: We have corrected it.
Comments 27:
Lines 451–454: Please rephrase the final sentence for clarity and consistency. Also, use “3rd generation” instead of “third generation”.
Response 27: We have modified it.
Comments 28:
Reference 16: Please provide the chapter title and page number.
Response 28: We have provided it.
Comments 29:
Reference 33: If the reviewer is not mistaken, this appears to be a publication by the same authors, based (at least partially) on the same dataset, published in 2023. Therefore, this reference is missing in the Material and Methods and would be appropriate in line 93, where origin of the samples is first mentioned, as well as in the rest of the methods.
Given the overlap with the previous publication, it is particularly important to clearly explain how the ISCAID guidelines were applied in this study, as this represents the main methodological difference and justifies the current work as a distinct contribution.
Response 29: The reviewer is right, and a specific part in the material and methods section focusing on the ISCAID guidelines application was added (lines 101-108), as well as in the discussion part (lines 463-468). However, compared with our previous paper that described and analyzed a general overview of AMR isolates within the veterinary hospital, this work was specifically focused on bacterial UTIs, their ISCAID classification and the temporal trend of MDR and antimicrobial use over a longer period.
Comments 30:
Reference 49: This is the wrong reference. Please correct.
Response 30: We have corrected it.
Reviewer 3 Report
Comments and Suggestions for Authors
Thank you for the opportunity to review this manuscript. It is a great topic and was quite enjoyable to read.
Simple Summary:
Line 20: I prefer not saying that fluoroquinolones are first-line antibiotics. Maybe you are referring to pyelonephritis here? Certainly, we would not use them first for simple/sporadic UTIs?
Introduction: Well written.
Materials and Methods:
Do we know what proportion of submitted samples came from primary care service vs internal medicine, emergency, or surgery? How often does your primary care and internal medicine submit urine culture on sporadic UTIs to confirm infection prior to treatment? Is this likely a skewed sample from being at a university teaching hospital and only seeing the cases that are complicated or didn’t respond or are more willing to submit and pay for a culture? This is a common bias in this type of study but should be mentioned and transparent as much as possible.
There is an implication that there was some effort to apply the ISCAID guidelines at the hospital over this timeframe and that might be the reason for the improvement over time, but there is no description in the methods about how clinicians were educated/encouraged to follow ISCAID guidelines. Can you please clarify?
Lines 98-99: Was route of collection recorded? Such as free-catch, cystocentesis, catheter? Did you exclude any samples collected by free catch (voided)?
Line 100: Was amount of growth recorded in CFU/ml? Perhaps expanding on “adequate bacterial growth” would be helpful. It is important to have criteria for minimum CFUs/ml considered a true UTI, typically based on route of urine collection, with at least 1000CFU/ml needed. Did you have higher CFU/ml criteria for voided samples, such as >100,000 (if allowed)?
Why disc diffusion instead of MIC?
Results:
I don’t think you need to include which bacterial species are statistically more common in cats vs dogs in both the text and Table 1. If you leave it in Table 1, please make it more clear in the Table heading what this p value is actually comparing “comparison of bacterial species between canine and feline isolates” or something similar.
Figure 1. I apologize, but I am really struggling to match up the colors with the drugs and make sense of this figure. Is there a way to include percentages and/or lines to connect them better? I don’t see amoxi-clav in the legend, even though the text says it is the biggest portion?
Figure 2. Again, there seem to be more color distributions in the dog bar than there are in the legend. It is challenging to follow and interpret this figure.
Table 4. Please clarify what the p value test is comparing…all 3 previous columns or just dogs vs cats?
Figure 3. If there are significant differences over time (from baseline?), please indicate those on the figure. Dashed lines should be described in the legend or title.
Lines 250-261: Perhaps you can explain in a user-friendly way what the Odds Ratio means in this context, seeing as they are less than 1?
Lines 273-276: is this for dogs or cats or both combined?
Discussion
Line 400: This difference between amoxi-clav and ampicillin has a nice similarity to the paper below which found 92% susceptibility in canine amoxiclav E.coli isolates, and only 53% susc to amoxicillin.
- https://onlinelibrary.wiley.com/doi/full/10.1111/jvim.15674
Line 408-409: This sentence is confusing to me as written. You say they are seldom used, but your justification is that they are broad spectrum and easy to administer? In my experience, those are typically reasons why many veterinarians do choose to use an antibiotic.
Author Response
Comments 1:
Simple Summary:
Line 20: I prefer not saying that fluoroquinolones are first-line antibiotics. Maybe you are referring to pyelonephritis here? Certainly, we would not use them first for simple/sporadic UTIs?
Response 1: The sentence was modified to understand that penicillins are first-line drugs for sporadic cystitis and fluoroquinolones are used firstly for pyelonephritis.
Comments 2: Introduction: Well written. Materials and Methods: Do we know what proportion of submitted samples came from primary care service vs internal medicine, emergency, or surgery? How often does your primary care and internal medicine submit urine culture on sporadic UTIs to confirm infection prior to treatment? Is this likely a skewed sample from being at a university teaching hospital and only seeing the cases that are complicated or didn’t respond or are more willing to submit and pay for a culture? This is a common bias in this type of study but should be mentioned and transparent as much as possible. Response 2: the reviewer is right, that is a common bias of this type of study and which implications were mentioned in the discussion (lines 485-489). Unfortunately, it was not possible to discriminate between the different services of the hospital from which the samples were taken.
|
Comments 3: There is an implication that there was some effort to apply the ISCAID guidelines at the hospital over this timeframe and that might be the reason for the improvement over time, but there is no description in the methods about how clinicians were educated/encouraged to follow ISCAID guidelines. Can you please clarify? Response 3: in the material and method section a sentence to clarify daily application of ISCAID guidelines was written as requested (lines 101-108). |
Comments 4:
Lines 98-99: Was route of collection recorded? Such as free-catch, cystocentesis, catheter? Did you exclude any samples collected by free catch (voided)?
Response 4: as mentioned in table A2, the specimens were taken by cystocentesis or catheterization. A clear classification was not performed since this information was not collected consistently, but we can anecdotally report that the vast majority (around 90%) was taken by cystocentesis.
Comments 5:
Line 100: Was amount of growth recorded in CFU/ml? Perhaps expanding on “adequate bacterial growth” would be helpful. It is important to have criteria for minimum CFUs/ml considered a true UTI, typically based on route of urine collection, with at least 1000CFU/ml needed. Did you have higher CFU/ml criteria for voided samples, such as >100,000 (if allowed)?
Response 5: The reviewer is right, but unfortunately, we did not record systematic data about the minimum CFU/ml to consider true UTIs. Nevertheless, the laboratory did not accept voided urine samples and processed only urines collected by cystocentesis and catheterization. The criteria used to consider a sample as positive considered the clinical findings and the bacteriological evaluation, including the sampling method (urines collected by catheterization are considered potentially more contaminated than urines collected by cystocentesis) and the animal species (healthy cats tend to have sterile urines) (Balboni et al, 2020 and 2024).
Balboni, A.; Franzo, G.; Bano, L.; Urbani, L.; Segatore, S.; Rizzardi, A.; Cordioli, B.; Cornaggia, M.; Terrusi, A.; Vasylyeva, K.; et al. No Viable Bacterial Communities Reside in the Urinary Bladder of Cats with Feline Idiopathic Cystitis. Research in Veterinary Science 2024, 168, 105137, doi:10.1016/j.rvsc.2024.105137.
Balboni, A.; Franzo, G.; Bano, L.; De Arcangeli, S.; Rizzardi, A.; Urbani, L.; Segatore, S.; Serafini, F.; Dondi, F.; Battilani, M. Culture-Dependent and Sequencing Methods Revealed the Absence of a Bacterial Community Residing in the Urine of Healthy Cats. Front. Vet. Sci. 2020, 7, 438, doi:10.3389/fvets.2020.00438.
Comments 6:
Why disc diffusion instead of MIC?
Response 6: Although MIC would have given more precise results, the reason behind the disc diffusion use is due to the resource limitations that the laboratory that performed the analysis presented.
Comments 7:
Results:
I don’t think you need to include which bacterial species are statistically more common in cats vs dogs in both the text and Table 1. If you leave it in Table 1, please make it more clear in the Table heading what this p value is actually comparing “comparison of bacterial species between canine and feline isolates” or something similar.
Response 7: the reviewer is right, and the Table heading was modified.
Comments 8:
Figure 1. I apologize, but I am really struggling to match up the colors with the drugs and make sense of this figure. Is there a way to include percentages and/or lines to connect them better? I don’t see amoxi-clav in the legend, even though the text says it is the biggest portion?
Figure 2. Again, there seem to be more color distributions in the dog bar than there are in the legend. It is challenging to follow and interpret this figure.
Response 8: The reviewer is right, the two figures presented mistakes in the colors that were corrected.
Comments 9:
Table 4. Please clarify what the p value test is comparing…all 3 previous columns or just dogs vs cats?
Response 9: the reviewer is right, and the Table heading was modified.
Comments 10:
Figure 3. If there are significant differences over time (from baseline?), please indicate those on the figure. Dashed lines should be described in the legend or title.
Response 10: the reviewer is right, and the Table heading was modified. The significant decrease over time from the baseline (1st semester) is shown through the tendency lines.
Comments 11:
Lines 250-261: Perhaps you can explain in a user-friendly way what the Odds Ratio means in this context, seeing as they are less than 1?
Response 11: in this context, an OR less than 1 indicates that increasing the semester of observation (from 1 to 5), the probability of MDR or single non-susceptibility is decreasing.
Comments 12:
Lines 273-276: is this for dogs or cats or both combined?
Response 12: This statement refers to both dogs and cats.
Comments 13:
Discussion
Line 400: This difference between amoxi-clav and ampicillin has a nice similarity to the paper below which found 92% susceptibility in canine amoxiclav E.coli isolates, and only 53% susc to amoxicillin.
Response 13: the reviewer is right, such similarity is worth citing and was added in the text (lines 440-442).
Comments 14:
Line 408-409: This sentence is confusing to me as written. You say they are seldom used, but your justification is that they are broad spectrum and easy to administer? In my experience, those are typically reasons why many veterinarians do choose to use an antibiotic.
Response 14: The reviewer is right, there was a mistake in the adverb used that was corrected.
Reviewer 4 Report
Comments and Suggestions for Authors
This study entitled “Antimicrobial Resistance in Companion Animals: A 30-months 2 Analysis on Clinical Isolates from Urinary Tract Infections in a 3 Veterinary Hospital” examined the prevalence of bacterial species, antimicrobial resistance patterns, and multi-drug resistance in clinical urinary tract infection isolates from dogs and cats at a European veterinary hospital from Italy and offered valuable insights regarding this pathology and antimicrobial resistance trends. The study emphasizes the importance of using international UTI guidelines for antimicrobial stewardship and highlights the role of bacteriological assessments in guiding effective, evidence-based antibiotic treatment to reduce resistance.
However, I do have several questions and suggestions for the authors.
The introduction section could be a little more detailed. Are there any data regarding UTIs in your region or country/Europe? Are there any recent studies worth mentioning which focused on UTIs in companion animals? What about other type of bacterial infections and AMR patterns compared to urinary tract infections?
Line 100 - Please explain what adequate bacterial growth stands for. Was this assessed using a quantitative method (number of CFU/mL)? If so, please briefly explain and mention the criteria used for the confirmation of UTIs depending on the method used for sampling (cystocentesis/catheterization).
Table 1 – Please check that all bacterial species names are written in italic (Streptococcus canis). Can you mention what other Enterobacter and Enterococcus species were identified?
Figure 1 – Please check the typos in the title and in the legend (aminoglycosides instead of aminoglicosides, doxycycline instead of doxicycline, clindamycin instead of clindamicyn). Same for Figure 2 (legend and title).
Table 4 – Check spelling for clindamycin
Line 310 – check spelling for S. pseudintermedius
Line 453 – Regarding One Health and zoonotic potential, why not discussing this in a more detailed manner in the discussion section? It is indeed, an important aspect to be taken into account. What bacterial species among the ones you identified are considered potentially zoonotic and may possess a threat to public health? Is there any published data regarding the transmission of these microorganisms to people?
Author Response
Comments 1:
This study entitled “Antimicrobial Resistance in Companion Animals: A 30-months 2 Analysis on Clinical Isolates from Urinary Tract Infections in a 3 Veterinary Hospital” examined the prevalence of bacterial species, antimicrobial resistance patterns, and multi-drug resistance in clinical urinary tract infection isolates from dogs and cats at a European veterinary hospital from Italy and offered valuable insights regarding this pathology and antimicrobial resistance trends. The study emphasizes the importance of using international UTI guidelines for antimicrobial stewardship and highlights the role of bacteriological assessments in guiding effective, evidence-based antibiotic treatment to reduce resistance.
However, I do have several questions and suggestions for the authors.
The introduction section could be a little more detailed. Are there any data regarding UTIs in your region or country/Europe? Are there any recent studies worth mentioning which focused on UTIs in companion animals? What about other type of bacterial infections and AMR patterns compared to urinary tract infections?
Response 1: The reviewer is right. Although a deeper analysis of similar data regarding UTI and companion animals is detailed in the discussion part, we added a sentence also in the introduction part (lines 82-85). The comparison of AMR rates between UTI and other bacterial infections was not included. Although it could be interesting, it would be beyond the scope of the study, in authors’ opinion
Comments 2:
Line 100 - Please explain what adequate bacterial growth stands for. Was this assessed using a quantitative method (number of CFU/mL)? If so, please briefly explain and mention the criteria used for the confirmation of UTIs depending on the method used for sampling (cystocentesis/catheterization).
Response 2: The determination of sample positivity was based on both a quantitative and a clinical evaluation. We did not measure systematically the number of CFU/ml as a single cut-off; instead, we combine the clinical findings with the bacteriological evaluation, that took into account the method used for sampling (e.g. urines collected by catheterization were considered potentially more contaminated) and the animal species (healthy cats tend to have sterile urines) (Balboni et al, 2020 and 2024).
Balboni, A.; Franzo, G.; Bano, L.; Urbani, L.; Segatore, S.; Rizzardi, A.; Cordioli, B.; Cornaggia, M.; Terrusi, A.; Vasylyeva, K.; et al. No Viable Bacterial Communities Reside in the Urinary Bladder of Cats with Feline Idiopathic Cystitis. Research in Veterinary Science 2024, 168, 105137, doi:10.1016/j.rvsc.2024.105137.
Balboni, A.; Franzo, G.; Bano, L.; De Arcangeli, S.; Rizzardi, A.; Urbani, L.; Segatore, S.; Serafini, F.; Dondi, F.; Battilani, M. Culture-Dependent and Sequencing Methods Revealed the Absence of a Bacterial Community Residing in the Urine of Healthy Cats. Front. Vet. Sci. 2020, 7, 438, doi:10.3389/fvets.2020.00438.
Comments 3:
Table 1 – Please check that all bacterial species names are written in italic (Streptococcus canis). Can you mention what other Enterobacter and Enterococcus species were identified?
Response 3: The reviewer is right, and the species of Enterobacter and Enterococcus were added in the caption of Table 1.
Comments 4:
Figure 1 – Please check the typos in the title and in the legend (aminoglycosides instead of aminoglicosides, doxycycline instead of doxicycline, clindamycin instead of clindamicyn). Same for Figure 2 (legend and title).
Response 4: the reviewer is right, and the figures were corrected.
Comments 5:
Table 4 – Check spelling for clindamycin
Response 5: We have checked it as requested.
Comments 6:
Line 310 – check spelling for S. pseudintermedius
Response 6: We have modified it.
Comments 7:
Line 453 – Regarding One Health and zoonotic potential, why not discussing this in a more detailed manner in the discussion section? It is indeed, an important aspect to be taken into account. What bacterial species among the ones you identified are considered potentially zoonotic and may possess a threat to public health? Is there any published data regarding the transmission of these microorganisms to people?
Response 7: The reviewer is right, the One Health aspect is worth discussing more deeply. Considering our results, the most important species with zoonotic potential is E.coli, especially for its high frequency and its resistance patterns often mediated by plasmids. A sentence with references to interspecies transmission data was added in lines 468-470).
Round 2
Reviewer 2 Report
Comments and Suggestions for Authors
The authors have addressed all suggested changes to the reviewers' satisfaction. Only two minor adjustments are still needed:
-
In Figure 1, the legend is still missing amoxicillin-clavulanic acid.
-
In the newly added Table 5 (which was included as requested), please delete the period in “n. of isolates” and italicize the “n” to maintain consistent formatting. Additionally, please insert a space after the genus names and correct the formatting of the last two entries.
Author Response
Dear reviewer,
thank you for your fast response. We addressed the two minor adjustments you highlighted.
Kind regards
Raffaele Scarpellini